**Data Availability Statement:** All relevant data are within the paper and its Supporting Information files.

# Association of obesity, visceral adiposity, and sarcopenia with an increased risk of metabolic syndrome: A retrospective study

**Su Hwan Kim**[1,2☯], **Hyoun Woo Kang**[1☯], **Ji Bong Jeong**[1]*, **Dong Seok Lee**[1,2], **Dong-Won Ahn**[1], **Ji Won Kim**[1], **Byeong Gwan Kim**[1], **Kook Lae Lee**[1], **Sohee Oh**[3], **Soon Ho Yoon**[4], **Sang Joon Park**[4]

**1** Department of Internal Medicine, Seoul National University College of Medicine, Seoul Metropolitan Government Seoul National University Boramae Medical Center, Seoul, Republic of Korea, **2** Health Care Center, Seoul Metropolitan Government Seoul National University Boramae Medical Center, Seoul, Republic of Korea, **3** Medical Research Collaborating Center, Seoul Metropolitan Government Seoul National University Boramae Medical Center, Seoul, Republic of Korea, **4** Department of Radiology, Seoul National University College of Medicine, Seoul National University Hospital, Seoul, Republic of Korea

☯ These authors contributed equally to this work.

\* jibjeong@gmail.com

## Abstract

### Aims

Metabolic syndrome (MS) is a global health problem associated with an increased risk of diabetes mellitus (DM), cardiovascular disease (CVD), and cancer. Body composition parameters, including obesity, visceral adiposity, and sarcopenia contribute to the development of MS and CVD. Previous studies have investigated the association of individual body composition parameters with MS. Studies analyzing the association between multiple body composition parameters and MS have been rare. We aimed to investigate the association between MS and multiple body composition parameters, including obesity, visceral adiposity, and sarcopenia.

### Methods

A total of 13,620 subjects who underwent voluntary routine checkups at the Health Care Center of our institution between October 2014 and December 2019 were enrolled. Only data from the first examination of subjects who underwent repeated checkups were included. Clinical and laboratory data were collected. Skeletal muscle mass and visceral fat area (VFA) were measured using bioelectrical impedance analysis. Appendicular skeletal muscle mass (ASM) was divided by body weight (in kg) and expressed as a percentage (calculated as, ASM% = ASM × 100/Weight). Data were compared between the groups based on obesity, VFA, and ASM%. Logistic regression analysis was performed to determine the risk of MS in each group.

### Results

Body mass index and VFA were significantly higher in subjects with MS than in those without MS. ASM% was significantly lower in subjects with MS than in those without MS.

**Funding:** The authors received no specific funding for this work.

**Competing interests:** The authors have declared that no competing interests exist.

Subjects with obesity, visceral adiposity, or sarcopenia had a higher prevalence of MS than those without. As the number of metabolic components increased from 0 to 5, we identified a decreasing trend of ASM% and an increasing trend of VFA and BMI (P for trend < 0.001 for all). In the paired analyses, all the three body composition parameters showed additive effects in predicting MS. In the logistic regression analysis, the three parameters were associated with an increased risk of MS after adjustment for age, sex, hypertension, DM, dyslipidemia, smoking, alcohol intake, and C-reactive protein.

### Conclusions

Obesity, visceral adiposity, and sarcopenia showed additive effects on MS prediction. Subjects with obesity, visceral adiposity, or sarcopenia were significantly associated with the increased risk of MS after adjustment for multiple confounders. Increasing skeletal muscle and reducing visceral fat may be strategies for the prevention or treatment of MS.

### Introduction

Metabolic syndrome (MS) has become a global health problem and is associated with an increased risk of diabetes mellitus (DM) [1]. MS also increases the risk of cardiovascular disease (CVD) [2,3], cancer, and mortality [4–6].

Visceral adiposity is considered to contribute to the development of MS [7–13]. Visceral adiposity is also known to be a risk factor for DM [14], CVD [15], non-alcoholic fatty liver disease (NAFLD) [16], reflux esophagitis [17], and cancer [18]. Waist circumference (WC) can be used to measure visceral adiposity. However, WC is only a surrogate of visceral adiposity and does not measure visceral adiposity precisely. Another limitation of WC measurement is its poor reproducibility [19]. Recent studies that measured visceral adiposity using bioelectrical impedance analysis (BIA) showed that visceral adiposity is associated with MS [8,10,11,13]. BIA can measure body fat and muscle mass easily and is cost-effective; thus, BIA is widely used.

It is known that obesity is related to MS, hypertension (HT), DM, dyslipidemia (DL), and CVD [20–24]. However, body mass index (BMI), which is an indicator of obesity, does not precisely reflect the amount of body fat and is limited in predicting obesity-related diseases [25].

Sarcopenia is the progressive loss of skeletal muscle mass [26,27]. With aging of the population globally, sarcopenia has become a global issue [28,29]. Loss of skeletal muscle mass is a known risk factor for MS [30–32], NAFLD [33,34], carotid atherosclerosis, and CVD [28,35,36]. Skeletal muscle is the main site of glucose uptake and utilization [37]; thus, sarcopenia increases insulin resistance and thereby induces DM and MS [32]. Sarcopenia limits physical activity and independent daily living [38]. Sarcopenia has been reported to increase morbidity [39], disability [40], medical costs [41], and mortality [42].

Many studies have investigated the association of individual body composition parameters with MS. However, few studies have analyzed the association between multiple body composition parameters and MS [43–45]. We aimed to investigate the association between MS and multiple body composition parameters, including obesity, visceral adiposity, and sarcopenia.

### Materials and methods

#### Study population

A total of 20,998 subjects underwent voluntary routine checkups at our Institutional Health Care Center between October 2014 and December 2019. After excluding 4,621 subjects who

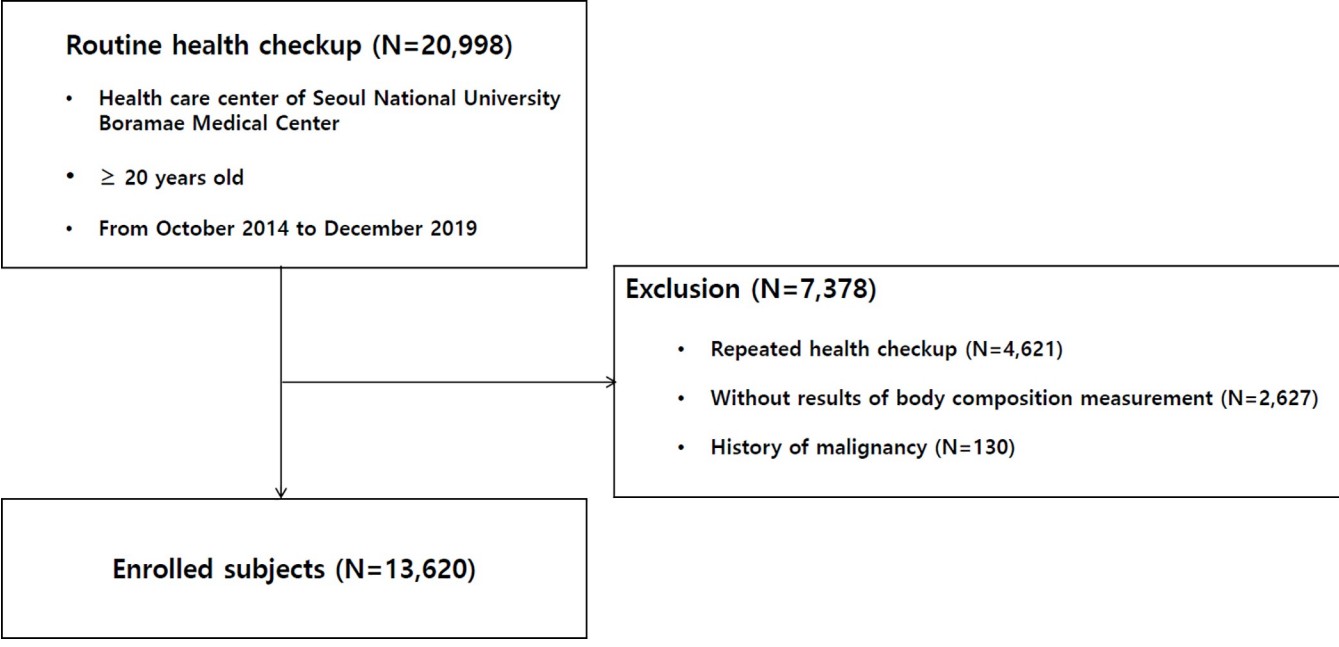

**Fig 1. Enrollment flow chart of patients.**

underwent repeated checkups, only data from the first examination were included. After excluding 2,627 subjects with insufficient data and 130 subjects with a history of malignancy, 13,620 subjects were enrolled, similar to a previous study (**Fig 1**) [46]. The data were fully anonymized before obtaining them. This study was approved by the Institutional Review Board of Boramae Medical Center (IRB No. 10-2020-234). The requirement for written informed consent was waived due to the retrospective nature of our study. The study was conducted in accordance with the Declaration of Helsinki.

## Data collection

The participants visited our health care center after an overnight 12-hour fast. Clinical information and blood laboratory data were collected during the health checkup. Height and weight were measured with the participants in a standing position wearing a light examination gown and no shoes. WC was measured at the umbilicus level with the participants in a standing position. Body composition analysis was performed using Inbody 720 (Biospace Co., Seoul, Korea) by a trained nurse following the manufacturer's protocol [47]. Using Inbody 720, skeletal muscle mass and visceral fat area (VFA) were automatically calculated. Clinical information was collected and included for the following parameters: age, sex, systolic and diastolic blood pressure (BP), smoking, alcohol intake, and medical history, including HT and DM. The following laboratory blood investigations were performed: total cholesterol, high-density lipoprotein cholesterol (HDL-C), low-density lipoprotein cholesterol (LDL-C), triglycerides (TG), glucose, aspartate aminotransferase (AST), alanine aminotransferase (ALT), uric acid, insulin level, and C-reactive protein (CRP).

## Definitions

BMI was defined as weight (in kg) divided by height squared (in $m^2$); obesity was defined as BMI $\geq$ 25 $kg/m^2$ based on the criteria for the Asia-Pacific region [45,48]. Subjects were divided into obese (BMI $\geq$ 25 $kg/m^2$) and non-obese (BMI < 25 $kg/m^2$) groups.

VFA was measured using Inbody 720 and used to assess visceral adiposity. Subjects with VFA ≥100 cm² were placed in the visceral adiposity group [11,43,49]. Subjects with VFA <100 cm² were placed in the group without visceral adiposity.

Appendicular skeletal muscle mass (ASM) was calculated as the sum of the lean skeletal muscle mass of the bilateral upper and lower limbs. ASM was divided by body weight (in kg) and expressed as a percentage (calculated as, ASM% = ASM × 100/Weight). Sarcopenia was defined as ASM% < 29.0 in males and < 22.9 in females [50,51]. Subjects were placed in the sarcopenia and non-sarcopenia groups, accordingly. Obesity, visceral adiposity, and sarcopenia were considered as prognostic body composition parameters.

HT was defined as systolic BP ≥ 140 mmHg, diastolic BP ≥ 90 mmHg, or the use of antihypertensive medications. DM was defined as fasting plasma glucose ≥ 126 mg/dL, glycated hemoglobin level ≥ 6.5%, or the use of anti-diabetic medications including insulin. DL was defined as TG level ≥ 150 mg/dL, HDL-C in males < 40 mg/dL and in females < 50 mg/dL, or the use of medications.

MS was defined when three or more of the following criteria were met: 1) WC in males ≥ 102 cm and in females ≥ 88 cm; 2) TG level ≥ 150 mg/dL or the use of medications; 3) HDL-C in males < 40 mg/dL and in females < 50 mg/dL, or the use of medications; 4) systolic BP ≥ 130 mmHg, diastolic BP ≥ 85 mmHg, or the use of antihypertensive medications; and 5) fasting plasma glucose ≥ 100 mg/dL or the use of anti-diabetic medications including insulin [52,53]. Severe MS was defined when four or more of the above criteria were met.

The homeostatic model assessment of insulin resistance (HOMA-IR) was calculated as [fasting glucose (mg/dL) × fasting insulin (μU/mL)]/405 [54].

## Comparison of Inbody 720 and computed tomography data

To verify the data of VFA and skeletal muscle mass as measured by Inbody 720, data were correlatively analyzed in subjects who underwent body composition analysis by BIA and computed tomography (CT) on the same day. Using CT, VFA and total abdominal muscle area (TAMA) were measured at the L3 vertebral level, which showed the highest correlation with visceral fat volume and whole-body skeletal muscle in previous studies [55,56].

All abdominal CT scans were performed using a 64-slice multi-detector CT scanner (Brilliance 64 scanners; Philips Healthcare, Amsterdam, Netherlands). Pre-contrast CT images were analyzed using a commercially available segmentation software program (MEDIP Deep Catch v1.0.0.0, MEDICALIP Co. Ltd., Seoul, South Korea) to measure TAMA. After automatic segmentation, the reader selected the level of the inferior endplate of the L3 vertebra and extracted the TAMA and VFA at the corresponding levels as previously described [46,57,58].

## Statistical analysis

Continuous variables were expressed as mean ± standard deviation. Categorical variables were presented as numbers and percentages. The Student's t-test and chi-square test were performed to compare quantitative and categorical variables, respectively. The linear trend between the number of MS components and categorical variables (obesity, visceral adiposity, and sarcopenia) was examined using the Cochran-Armitage trend test. The linear trend between the number of MS components and continuous variables (BMI, VFA, ASM%) was examined using analysis of variance with linear contrast. Logistic regression analysis was performed to determine the risk of MS. Crude odds ratios (ORs) were calculated for obesity, visceral adiposity, and sarcopenia at baseline. Model 1 was adjusted for age and sex; model 2 was adjusted for age, sex, HT, DM, and DL; model 3 was adjusted for age, sex, HT, DM, DL, smoking, and alcohol intake; and model 4 was adjusted for age, sex, HT, DM, DL, smoking, alcohol

intake, and CRP levels. Pearson correlation analysis was performed between the BIA and CT scan data. Statistical significance was set at P < 0.05. All statistical analyses were conducted using the IBM SPSS 26 statistical software (IBM Corp., Armonk, NY, USA).

## Results

### Baseline characteristics of the study population

Among the 13,620 subjects who underwent routine health checkups, 7,422 and 6,198 were males and females, respectively. A total of 2,238 subjects were diagnosed with MS. Clinical characteristics according to the presence of MS are presented in **Table 1** [46]. Clinical and anthropometric characteristics were significantly different based on the presence of MS. BMI and VFA were significantly higher in subjects with MS than in those without MS. ASM% was significantly lower in subjects with MS than that in those without MS.

Table 1. Clinical characteristics according to metabolic syndrome [46].

| Variables | All (N = 13620) | | p value |
|---|---|---|---|
| | No metabolic syndrome | Metabolic syndrome | |
| | N = 11382 (83.6%) | N = 2238 (16.4%) | |
| Age (years) | 46.79±12.85 | 54.91±12.20 | <0.001 |
| Weight (kg) | 64.33±11.91 | 73.62±14.80 | <0.001 |
| BMI (kg/m$^2$) | 23.13±3.03 | 26.62±3.75 | <0.001 |
| WC (cm) | 81.94±8.94 | 91.83±9.36 | <0.001 |
| Systolic BP (mmHg) | 115.35±14.95 | 128.26±15.35 | <0.001 |
| Diastolic BP (mmHg) | 78.04±10.57 | 84.48±11.42 | <0.001 |
| Visceral fat area (cm$^2$) | 85.75±31.48 | 120.65±38.49 | <0.001 |
| ASM (kg) | 19.58±4.75 | 20.96±5.21 | <0.001 |
| ASM% | 30.25±3.55 | 28.33±3.56 | <0.001 |
| Cholesterol (mg/dL) | 196.87±34.79 | 193.76±42.60 | 0.001 |
| HDL-C (mg/dL) | 58.45±13.96 | 45.97±10.94 | <0.001 |
| LDL-C (mg/dL) | 119.15±32.71 | 113.18±37.60 | <0.001 |
| Triglyceride (mg/dL) | 96.88±59.40 | 179.16±111.09 | <0.001 |
| Glucose (mg/dL) | 90.70±15.17 | 113.48±28.63 | <0.001 |
| AST (IU/L) | 26.48±17.52 | 33.58±20.49 | <0.001 |
| ALT (IU/L) | 25.49±22.45 | 39.10±31.98 | <0.001 |
| Uric acid (mg/dL) | 5.17±1.29 | 5.64±1.47 | <0.001 |
| HbA1c (%) | 5.50±0.54 | 6.24±1.10 | <0.001 |
| Insulin | 8.55±3.87 | 13.42±8.87 | <0.001 |
| HOMA-IR | 2.02±1.11 | 3.80±2.33 | <0.001 |
| C-reactive protein (mg/dL) | 0.13±0.44 | 0.22±0.55 | <0.001 |
| Hypertension (%) | 2594 (22.8) | 1636 (73.1) | <0.001 |
| Diabetes mellitus (%) | 433 (3.8) | 706 (31.5) | <0.001 |
| Smoking (%) | 1897 (16.7) | 484 (21.6) | <0.001 |
| Alcohol intake (%) | 6096 (53.6) | 1128 (50.4) | 0.006 |

Values are presented as mean ± standard deviation (SD) or number (%).

BMI, body mass index; WC, waist circumference; BP, blood pressure.

ASM, appendicular skeletal muscle; ASM%, appendicular skeletal muscle percentage.

HDL-C, high-density lipoprotein cholesterol; LDL-C, low-density lipoprotein cholesterol.

AST, aspartate aminotransferase; ALT, alanine aminotransferase.

HOMA-IR, homeostatic model assessment of insulin resistance.

**Table 2. Clinical characteristics according to obesity, visceral adiposity, and sarcopenia [46].**

| Variables | All (N = 13620) | | p value | All (N = 13620) | | p value | All (N = 13620) | | p value |
|---|---|---|---|---|---|---|---|---|---|
| | Non-obese | Obesity | | None | Visceral adiposity | | None | Sarcopenia | |
| | N = 9164 (67.3%) | N = 4456 (32.7%) | | N = 8657 (63.6%) | N = 4963 (36.4%) | | N = 12554 (92.2%) | N = 1066 (7.8%) | |
| Age (years) | 47.57±13.34 | 49.28±12.51 | <0.001 | 46.65±13.19 | 50.70±12.53 | <0.001 | 47.70±12.84 | 53.16±14.94 | <0.001 |
| Weight (kg) | 60.03±9.05 | 77.85±11.23 | <0.001 | 60.20±9.65 | 75.73±11.85 | <0.001 | 64.85±11.96 | 77.73±16.92 | <0.001 |
| BMI (kg/m$^2$) | 21.87±2.01 | 27.48±2.50 | <0.001 | 22.17±2.51 | 26.39±3.12 | <0.001 | 23.29±2.99 | 28.67±4.11 | <0.001 |
| WC (cm) | 79.04±7.22 | 92.88±7.29 | <0.001 | 79.12±7.68 | 91.34±7.86 | <0.001 | 82.49±8.90 | 96.30±10.05 | <0.001 |
| Systolic BP (mmHg) | 113.69±14.77 | 125.25±14.86 | <0.001 | 113.12±14.59 | 125.06±14.83 | <0.001 | 116.66±15.50 | 127.09±15.64 | <0.001 |
| Diastolic BP (mmHg) | 76.86±10.31 | 83.70±10.87 | <0.001 | 76.56±10.23 | 83.53±10.82 | <0.001 | 78.69±10.82 | 83.87±11.69 | <0.001 |
| Visceral fat area (cm$^2$) | 77.04±25.52 | 121.19±33.71 | <0.001 | 70.69±18.47 | 127.76±27.10 | <0.001 | 87.18±30.75 | 142.13±43.73 | <0.001 |
| ASM (kg) | 18.44±4.35 | 22.62±4.62 | <0.001 | 18.35±4.36 | 22.35±4.61 | <0.001 | 19.76±4.80 | 20.43±5.38 | <0.001 |
| ASM% | 30.44±3.64 | 28.90±3.36 | <0.001 | 30.24±3.64 | 29.40±3.54 | <0.001 | 30.26±3.49 | 26.07±2.90 | <0.001 |
| Cholesterol (mg/dL) | 194.62±35.31 | 199.93±37.73 | <0.001 | 193.69±34.69 | 201.02±38.26 | <0.001 | 196.03±35.87 | 200.18±39.76 | 0.001 |
| HDL-C (mg/dL) | 59.68±14.46 | 49.65±11.18 | <0.001 | 59.91±14.54 | 50.27±11.48 | <0.001 | 57.03±14.30 | 49.01±11.66 | <0.001 |
| LDL-C (mg/dL) | 116.13±32.69 | 122.53±35.05 | <0.001 | 115.36±32.16 | 123.22±35.47 | <0.001 | 117.81±33.26 | 122.86±37.16 | <0.001 |
| Triglyceride (mg/dL) | 94.63±57.76 | 142.83±98.12 | <0.001 | 92.69±57.60 | 141.30±94.43 | <0.001 | 107.35±74.49 | 146.33±93.26 | <0.001 |
| Glucose (mg/dL) | 91.64±17.76 | 100.22±22.79 | <0.001 | 90.81±16.23 | 100.77±23.89 | <0.001 | 93.66±19.16 | 103.61±25.96 | <0.001 |
| AST (IU/L) | 25.82±16.37 | 31.41±21.06 | <0.001 | 25.44±16.91 | 31.50±19.76 | <0.001 | 27.01±17.79 | 35.13±21.44 | <0.001 |
| ALT (IU/L) | 23.07±20.22 | 37.32±30.02 | <0.001 | 22.45±20.88 | 36.93±28.16 | <0.001 | 26.37±22.94 | 43.74±37.15 | <0.001 |
| Uric acid (mg/dL) | 4.98±1.24 | 5.80±1.36 | <0.001 | 4.94±1.24 | 5.78±1.33 | <0.001 | 5.19±1.31 | 5.94±1.48 | <0.001 |
| HbA1c (%) | 5.54±0.63 | 5.81±0.84 | <0.001 | 5.51±0.58 | 5.83±0.86 | <0.001 | 5.59±0.68 | 5.99±0.99 | <0.001 |
| Insulin | 7.94±3.39 | 12.14±7.43 | <0.001 | 7.70±3.14 | 12.18±7.26 | <0.001 | 8.72±3.82 | 16.85±11.09 | <0.001 |
| HOMA-IR | 1.95±1.22 | 3.11±1.93 | <0.001 | 1.83±1.02 | 3.19±1.96 | <0.001 | 2.17±1.22 | 4.37±2.84 | <0.001 |
| C-reactive protein (mg/dL) | 0.13±0.48 | 0.19±0.41 | <0.001 | 0.12±0.46 | 0.19±0.46 | <0.001 | 0.14±0.46 | 0.28±0.49 | <0.001 |
| Metabolic syndrome (%) | 770 (8.4) | 1468 (32.9) | <0.001 | 685 (7.9) | 1553 (31.3) | <0.001 | 1746 (13.9) | 492 (46.2) | <0.001 |
| Hypertension (%) | 2025 (22.1) | 2205 (49.5) | <0.001 | 1774 (20.5) | 2456 (49.5) | <0.001 | 3591 (28.6) | 639 (59.9) | <0.001 |
| Diabetes mellitus (%) | 574 (6.3) | 565 (12.7) | <0.001 | 465 (5.4) | 674 (13.6) | <0.001 | 947 (7.5) | 192 (18.0) | <0.001 |
| Smoking (%) | 1367 (14.9) | 1014 (22.8) | <0.001 | 1071 (12.4) | 1310 (26.4) | <0.001 | 2144 (17.1) | 237 (22.2) | <0.001 |
| Alcohol intake (%) | 4637 (50.6) | 2587 (58.1) | <0.001 | 4250 (49.1) | 2974 (59.9) | <0.001 | 6672 (53.1) | 552 (51.8) | 0.392 |

Values are presented as mean ± standard deviation (SD) or number (%).

BMI, body mass index; WC, waist circumference; BP, blood pressure; ASM, appendicular skeletal muscle.

ASM%, appendicular skeletal muscle percentage; HDL-C, high-density lipoprotein cholesterol; LDL-C, low-density lipoprotein cholesterol.

AST, aspartate aminotransferase; ALT, alanine aminotransferase; HOMA-IR, homeostatic model assessment of insulin resistance.

The subjects were divided into two groups according to obesity, VFA, and ASM% (Table 2) [46]. Subjects with obesity, visceral adiposity, or sarcopenia were significantly older and had higher LDL-C, TG, CRP, and glucose levels, and HOMA-IR than those without. Subjects with obesity, visceral adiposity, or sarcopenia had a higher prevalence of HT, DM, and MS than those without.

## Metabolic parameters according to obesity, visceral adiposity, and sarcopenia

Table 3 shows the trend in the number of metabolic parameters between subjects with and those without obesity, visceral adiposity, or sarcopenia. The prevalence of obesity, visceral adiposity, or sarcopenia increased with the number of metabolic parameters (P < 0.001 for all).

**Table 3. Comparisons of number of metabolic parameters according to obesity, visceral adiposity, or sarcopenia.**

| | | Obesity | | P-value | Visceral adiposity | | P-value | Sarcopenia | | P-value |
|---|---|---|---|---|---|---|---|---|---|---|
| | | No | Yes | | No | Yes | | No | Yes | |
| Number of parameters | N | | | <0.001 | | | <0.001 | | | <0.001 |
| 0 | 4891 | 4343 (88.8) | 548 (11.2) | | 4233 (86.5) | 658 (13.5) | | 4819 (98.5) | 72 (1.5) | |
| 1 | 3797 | 2636 (69.4) | 1161 (30.6) | | 2458 (64.7) | 1339 (35.3) | | 3584 (94.4) | 213 (5.6) | |
| 2 | 2694 | 1415 (52.5) | 1279 (47.5) | | 1281 (47.6) | 1413 (52.4) | | 2405 (89.3) | 289 (10.7) | |
| 3 | 1513 | 607 (40.1) | 906 (59.9) | | 537 (35.5) | 976 (64.5) | | 1235 (81.6) | 278 (18.4) | |
| 4 | 605 | 158 (26.1) | 447 (73.9) | | 139 (23.0) | 466 (77.0) | | 446 (73.7) | 159 (26.3) | |
| 5 | 120 | 5 (4.2) | 115 (95.8) | | 9 (7.5) | 111 (92.5) | | 65 (54.2) | 55 (45.8) | |

Fig 2A shows the differences in the prevalence of MS between subjects with and without obesity, visceral adiposity, or sarcopenia. The prevalence of MS was significantly higher in those with obesity, visceral adiposity, or sarcopenia than in those without (32.9% vs. 8.4%, 31.3% vs. 7.9%, and 46.2% vs. 13.9%, respectively, P < 0.001 for all). We also calculated the prevalence of severe MS based on the presence of either four or five criteria among the groups (Fig 2B). The prevalence of severe MS was significantly higher in those with obesity, visceral adiposity, or sarcopenia than in those without (12.6% vs. 1.8%, 11.6% vs. 1.7%, and 20.1% vs. 4.1%, respectively, P < 0.001 for all).

The descriptive numerical values of BMI, VFA, and ASM% showed linear trends according to the number of metabolic parameters (Table 4). As the number of metabolic components increased from 0 to 5, a decreasing trend of ASM% and an increasing trend of BMI and VFA were identified (P < 0.001 for all).

## Additive effects of body composition parameters on metabolic syndrome

The number of metabolic parameters was calculated according to the numbers of prognostic body composition parameters, including obesity, visceral adiposity, and sarcopenia. Metabolic burden increased with an increase in the number of body composition parameters (P for trend < 0.001, Fig 3A). As the number of body composition parameters increased from 0 to 3, the prevalence of MS increased accordingly (5.9%, 18.0%, 33.6%, and 50.4%, respectively; P for

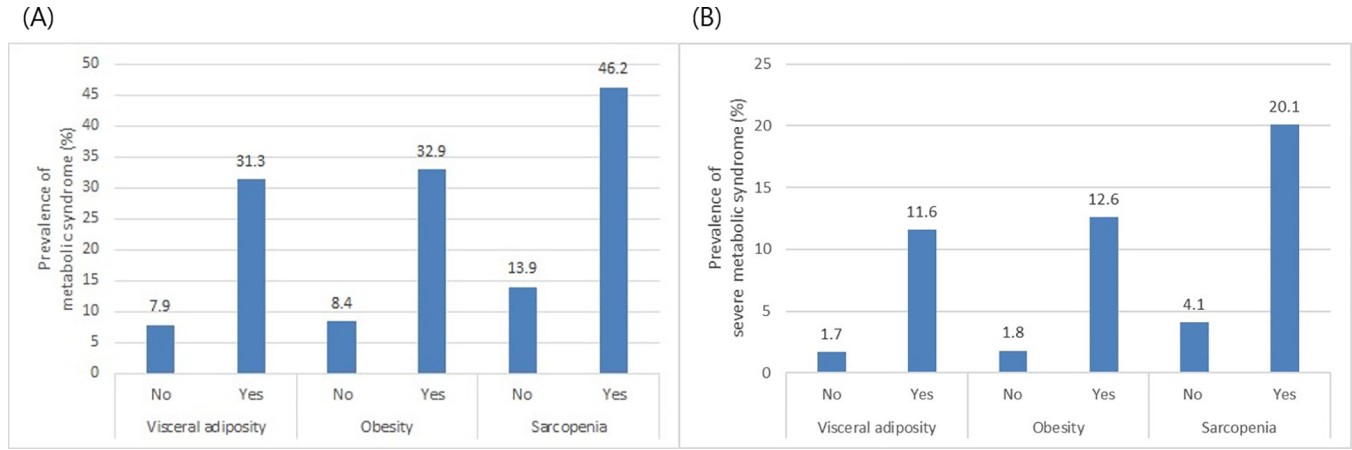

**Fig 2.** (A) The prevalence of MS in subjects with obesity, visceral adiposity, or sarcopenia was higher than that in subjects without these parameters (P < 0.001 for all). (B) The prevalence of severe MS in subjects with obesity, visceral adiposity, or sarcopenia was higher than that in subjects without them (P < 0.001 for all). MS, metabolic syndrome.

**Table 4. Descriptive numerical values of body mass index, visceral fat area, and ASM% according to the number of metabolic parameters.**

| Number of parameters | 0 | 1 | 2 | 3 | 4 | 5 | P-value |
|---|---|---|---|---|---|---|---|
| N | 4891 | 3797 | 2694 | 1513 | 605 | 120 | |
| BMI (kg/m$^2$) | 21.79±2.52 | 23.58±2.79 | 24.94±3.09 | 25.99±3.32 | 27.48±3.89 | 30.24±5.02 | <0.001 |
| Visceral fat area (cm$^2$) | 71.04±25.71 | 90.94±28.82 | 105.15±31.78 | 115.94±36.02 | 127.85±41.27 | 143.67±40.47 | <0.001 |
| ASM% | 30.49±3.56 | 30.30±3.56 | 29.75±3.49 | 28.91±3.47 | 27.55±3.39 | 24.88±2.68 | <0.001 |

BMI, body mass index; ASM%, appendicular skeletal muscle percentage.

P-value estimated from analysis of variance with linear contrast.

trend < 0.001 in all; **Fig 3B**) On performing paired analyses between two body composition parameters, all three parameters, including obesity, visceral adiposity, and sarcopenia, showed additive effects in predicting MS (**Fig 4A–4C**).

## Body composition parameters and metabolic syndrome

Obesity, visceral adiposity, and sarcopenia were significantly associated with the risk of MS (crude OR = 5.356, 5.300, and 5.306, respectively). After adjustment for age, sex, HT, DM, DL, smoking, alcohol intake, and CRP, ORs remained significant for obesity, visceral adiposity, and sarcopenia (adjusted OR = 4.235, 3.552, and 3.674, respectively; **Table 5**).

## Correlation of VFA and skeletal muscle mass between Inbody 720 and computed tomography

Among the enrolled subjects, CT scans were performed on 966 subjects on the same day as the Inbody 720 analysis. Thus, correlation analysis was conducted in 966 subjects, similar to a previous study [46]. VFA measured by Inbody 720 positively correlated with VFA measured by CT scan (R = 0.743, P < 0.001, **Fig 5A**). ASM measured using BIA was positively correlated with TAMA measured by CT scan (R = 0.890, P < 0.001, **Fig 5B**) as previously described [46].

## Discussion

In the current study, we analyzed the association between body composition parameters (obesity, visceral adiposity, and sarcopenia) and MS. We demonstrated that obesity, visceral

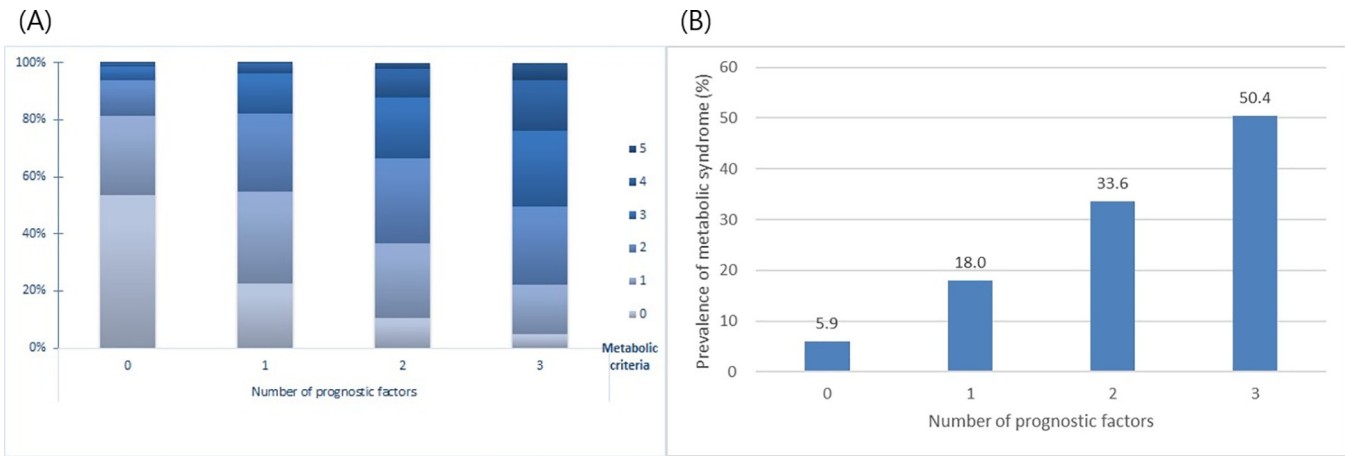

**Fig 3.** (A) The number of metabolic parameters according to the number of unfavorable body composition parameters (obesity, visceral adiposity, and sarcopenia). (B) The prevalence of MS according to the number of unfavorable body composition parameters (obesity, visceral adiposity, and sarcopenia). (P for trend < 0.001 in all); MS, metabolic syndrome.

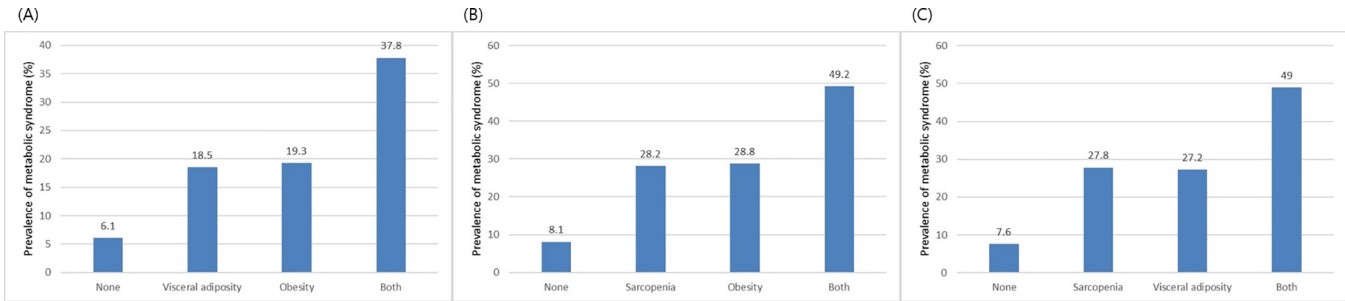

**Fig 4.** The prevalence of MS in paired analyses according to (A) obesity and visceral adiposity, (B) obesity and sarcopenia, and (C) visceral adiposity and sarcopenia; MS, metabolic syndrome.

adiposity, and sarcopenia were significantly associated with MS. After adjusting for multiple confounders, including age, sex, HT, DM, DL, smoking, alcohol intake, and CRP levels, subjects with obesity, visceral adiposity, and sarcopenia were found to be associated with an increased risk for MS (**Table 5**). In addition, as the number of prognostic body composition parameters increased, the risk for MS additively increased (**Figs 3 and 4**). Our study shows an association between body composition parameters and the risk of MS in a healthy population who underwent routine health checkups.

To the best of our knowledge, this is the largest study showing that unfavorable body composition parameters (obesity, visceral adiposity, and sarcopenia) additively increase the risk for MS. Previous studies have shown that visceral adiposity [7–11], sarcopenia [30–32], and obesity [20–23] were associated with an increased risk for MS. However, most of these studies analyzed the association of only a single body composition parameter with MS; studies analyzing the association between multiple body composition parameters and MS have been rare [43,45]. Lim et al. analyzed the association between MS and two parameters (sarcopenia and visceral adiposity). However, in the above-mentioned study, visceral adiposity was defined based on abdominal CT results. Moreover, ASM was measured by dual energy X-ray absorptiometry and the sample size was small (N = 565) [43]. Lu et al. analyzed the association between MS and two parameters (sarcopenia and obesity). In this study, BIA was utilized for the measurement of skeletal muscle mass; however, the sample size was also small (N = 600) [45].

The previous study of the same study population as this study (Fig 1) showed that sarcopenia diagnosed by BIA is independently associated with MS risk in a dose-response manner

**Table 5. Odds ratios of metabolic syndrome according to the presence of obesity, visceral adiposity, and sarcopenia.**

| | Obesity | | | Visceral adiposity | | | Sarcopenia | | |
|---|---|---|---|---|---|---|---|---|---|
| | OR | 95% CI | p value | OR | 95% CI | p value | OR | 95% CI | p value |
| Crude | 5.356 | 4.862–5.900 | <0.001 | 5.300 | 4.803–5.849 | <0.001 | 5.306 | 4.656–6.046 | <0.001 |
| Model 1 | 5.637 | 5.084–6.249 | <0.001 | 5.090 | 4.577–5.661 | <0.001 | 4.414 | 3.847–5.065 | <0.001 |
| Model 2 | 4.259 | 3.704–4.898 | <0.001 | 3.547 | 3.079–4.086 | <0.001 | 3.774 | 3.111–4.577 | <0.001 |
| Model 3 | 4.260 | 3.705–4.899 | <0.001 | 3.567 | 3.095–4.110 | <0.001 | 3.778 | 3.114–4.583 | <0.001 |
| Model 4 | 4.235 | 3.682–4.872 | <0.001 | 3.552 | 3.082–4.095 | <0.001 | 3.674 | 3.027–4.460 | <0.001 |

Model 1: Adjusted for age and sex.

Model 2: Adjusted for age, sex, HT, DM, and DL.

Model 3: Adjusted for age, sex, HT, DM, DL, smoking, and alcohol intake.

Model 4: Adjusted for age, sex, HT, DM, DL, smoking, alcohol intake, and CRP levels.

HT, hypertension; DM, diabetes mellitus; DL, dyslipidemia; CRP, C-reactive protein; OR, odds ratio; CI, confidence interval.

[46]. Our previous study focused on the relationship between sarcopenia and MS [46]. In contrast, the current study comprehensively analyzed the effects of obesity, visceral adiposity, and sarcopenia on the risk of MS. Our current study demonstrated that the risks for MS additively increased as the number of undesirable body composition parameters (obesity, visceral adiposity, and sarcopenia) increased from 0 to 3. By utilizing BIA for the measurements of skeletal muscle mass and VFA, as well as BMI, the population with a high risk for MS could be easily identified. Our study also showed that all three body composition parameters (obesity, visceral adiposity, and sarcopenia) were associated with increased MS risk after adjustments for age, sex, HT, DM, DL, smoking, alcohol intake, and CRP levels. In accordance with previous studies analyzing CRP and MS [59,60], CRP was adopted as a variable in our current study. In our study, as the number of metabolic components increased from 0 to 5, a decreasing trend of ASM% and an increasing trend of VFA and BMI were identified, which exhibited a dose-response manner.

Mechanisms that link sarcopenia and MS include insulin resistance and inflammation [61,62]. The skeletal muscle is the primary site of glucose utilization [37]; the role of sarcopenia in causing insulin resistance and DM has been described [32,63]. In our current study, HOMA-IR was significantly higher in the sarcopenia group than in the non-sarcopenic group. Another mechanism that links sarcopenia and MS is inflammation. The association between inflammation and sarcopenia has been reported [64,65]. Pro-inflammatory cytokines, such as interleukin (IL)-6 and tumor necrosis factor (TNF)-alpha, are associated with sarcopenia. In our current study, the sarcopenia group had higher CRP levels than the non-sarcopenic group, which supported the hypothesis that systemic inflammation serves as a link between sarcopenia and MS.

A link has been proposed between adipose tissue and skeletal muscle inflammation [66]. According to this mechanism, there is a positive feedback loop between visceral adiposity and sarcopenia. In obese subjects, adipose tissue is infiltrated by activated pro-inflammatory macrophages and is associated with an elevated production of pro-inflammatory molecules and adipokines [67,68]. The production of TNF-alpha, IL-6, and CRP from adipose tissue influences insulin resistance [69]. In our current study, the visceral adiposity group had higher CRP levels than the group without visceral adiposity, which supported the hypothesis that systemic inflammation serves as a link between visceral adiposity and MS.

CT has been considered the gold standard for measuring skeletal muscle mass [70]. Cross-sectional CT images of the lumbar skeletal muscle have provided good estimates of the total body skeletal muscle [71,72]. However, the recent use of CT for measuring body fat or muscle has been limited due to an increased risk for radiation exposure [11]. In our current study, BIA was used to measure ASM and VFA because BIA has been widely used owing to its accessibility, safety, and cost-efficiency [10,45,73,74]. Furthermore, to validate the data of VFA and skeletal muscle mass measured by Inbody 720, the correlation between BIA data and CT scans was analyzed in subjects who underwent body composition analysis using BIA and CT scans on the same day. VFA and TAMA at the L3 vertebral level measured by CT scan show a high correlation with visceral fat volume and whole-body skeletal muscle [55,56]. Recent studies reported that BIA-measured VFA indicated an increased risk for MS as precisely as CT-measured VFA [47,75]. Our study also showed a high positive correlation between BIA-measured ASM and CT-measured TAMA as previously described [46]. Moreover, a high correlation was also observed between BIA- and CT- measured VFA in our study (**Fig 5**).

In the current study, we used BIA to measure ASM and VFA similar to previous studies. However, the study size was relatively small in most of the earlier studies (N<1,500) [8,10,13,31,45]. Jeon et al. analyzed the risk for MS according to BIA-measured VFA; the study size was large, but skeletal muscle mass was not addressed [11]. The strength of our study is

(A) (B)

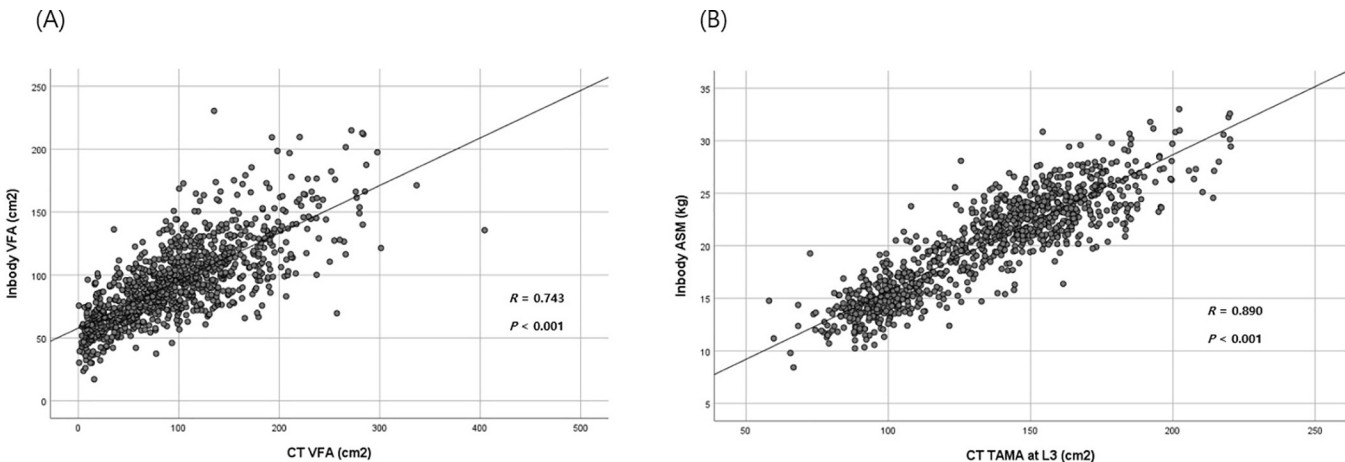

**Fig 5.** (A) Correlation of VFA measured by Inbody 720 and CT scan (R = 0.743, P<0.001), (B) Correlation of ASM measured by Inbody 720 and TAMA at the L3 vertebral level measured by CT scan (R = 0.890, P < 0.001) [46]. VFA, visceral fat area; CT, computed tomography; ASM, appendicular skeletal muscle mass; TAMA, total abdominal muscle area.

that we analyzed multiple body composition parameters, including VFA and ASM, in a large and healthy population. In addition, a significant association between body composition parameters and MS was confirmed after multivariable adjustments for age, sex, underlying diseases, smoking, alcohol intake, and inflammatory markers.

Our study had some limitations. First, as this was a cross-sectional, single-centered, retrospective study, the duration of MS was not assessed. It was difficult to assess the causal relationship between visceral fat, sarcopenia, and MS. Second, our study population was of an Asian ethnicity; thus, our study results cannot be generalized for all ethnicities. Third, our study population consisted of healthy subjects who underwent routine health checkups in a health care center. Thus, our study results are difficult to apply in non-healthy subjects. On the contrary, our study results are likely to be generalizable to a healthy population. Fourth, functional measurements of skeletal muscle, including handgrip testing or gait speed, were not performed. Surveys of exercise status were not included in the variables and could not be evaluated in our study. Finally, HOMA-IR results were available in only a small portion of the study population (N = 305); thus, significant results after adjustment for HOMA-IR could not be derived.

In conclusion, our study demonstrated that obesity, visceral adiposity, and sarcopenia were significantly associated with MS. In addition, with the increase in unfavorable body composition parameters, there is an additive increased risk of MS. Increasing skeletal muscle and reducing visceral adiposity may act as strategies for the prevention or treatment of MS. Further studies are needed to assess the causal relationship between body composition parameters and MS.

## Supporting information

**S1 File. Data file.**
(XLSX)

## Author Contributions

**Conceptualization:** Ji Bong Jeong.

**Data curation:** Su Hwan Kim, Hyoun Woo Kang, Soon Ho Yoon, Sang Joon Park.

**Formal analysis:** Su Hwan Kim, Hyoun Woo Kang, Sohee Oh.

**Methodology:** Soon Ho Yoon, Sang Joon Park.

**Supervision:** Ji Bong Jeong.

**Writing – original draft:** Su Hwan Kim, Hyoun Woo Kang.

**Writing – review & editing:** Su Hwan Kim, Hyoun Woo Kang, Ji Bong Jeong, Dong Seok Lee, Dong-Won Ahn, Ji Won Kim, Byeong Gwan Kim, Kook Lae Lee, Sohee Oh, Soon Ho Yoon, Sang Joon Park.

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
