## [Decision Letter · Decision Letter 0]

23 Jun 2021

PONE-D-21-15211

Association of obesity, visceral adiposity, and sarcopenia with an increased risk of metabolic syndrome: a retrospective study

PLOS ONE

Dear Dr. Jeong,

Thank you for submitting your manuscript to PLOS ONE. After careful consideration, we feel that it has merit but does not fully meet PLOS ONE’s publication criteria as it currently stands. Therefore, we invite you to submit a revised version of the manuscript that addresses the points raised during the review process.

We look forward to receiving your revised manuscript.

Kind regards,

Mauro Lombardo

Academic Editor

PLOS ONE

Journal Requirements:

2. Please ensure you have described your statistical analysis in sufficient detail, including any post-hoc analyses performed

Additional Editor Comments (if provided):

Dear authors

Please improve the English in the paper.

Answer point-by-point to the reviews' observations

Reviewers' comments:

Reviewer's Responses to Questions

**Comments to the Author**

1. Is the manuscript technically sound, and do the data support the conclusions?

Reviewer #1: Yes

Reviewer #2: Yes

2. Has the statistical analysis been performed appropriately and rigorously? 

Reviewer #1: Yes

Reviewer #2: Yes

3. Have the authors made all data underlying the findings in their manuscript fully available?

Reviewer #1: Yes

Reviewer #2: Yes

4. Is the manuscript presented in an intelligible fashion and written in standard English?

Reviewer #1: Yes

Reviewer #2: Yes

5. Review Comments to the Author

Reviewer #1: The manuscript provides evidence of the association between body composition parameters (obesity, visceral adiposity, and sarcopenia) and metabolic syndrome in a large sample of subjects. The manuscript is technically sound and the data do support the conclusions however there are a few revisions or clarifications I would suggest the authors address.

Statistical Analysis/Study Population (Methods General):

•The authors indicated that data from 4,621 repeated checkups were excluded from the analyses it would have been interesting to assess how the body composition parameters may have predicted changes in MS status over time using the repeated visits from the 4,621 with repeat data as this data was available.

Results:

•The results are clear however, the authors should include the percentages for the total number of subjects (n=2238) with MS as well as the percentages for the total number of subjects who were in the obesity group, visceral adiposity group and sarcopenia group. I would further suggest either updating this in Table 1 and Table 2.

•Table 3 and the language describing table 3 is unclear, it states the prevalence of MS was significantly higher in those with obesity, visceral adiposity AND sarcopenia compared to those without however, the table appears to show these as mutually exclusive groups. That is to say, obesity vs none, visceral vs non, sarcopenia vs none NOT those with obesity, visceral adiposity AND sarcopenia I would assume the group of individuals whom may be categorized by all 3 conditions would be much smaller. Thus, I would suggest the authors use ‘or’ rather than ‘and’ to indicate these analyses were done separately for each of the 3 conditions.

•It is unclear which models were used to assess the results reported in Table 3 and 4—were these done using chi-squared or the students t-tests? As these are counts for the number of parameters the authors should be more clear on how these results were modeled in the statistical analyses section.

•Table 5 is clearly defined and easy to read as well as reported and described appropriately both in the analyses section as well as the results.

Overall, the manuscript is clear and describes the association between obesity, visceral adiposity, and sarcopenia in a large sample of individuals.

Reviewer #2: Comment for Authors

The authors have undertaken study to explore the association of multiple body composition parameters such as obesity, visceral adiposity and sarcopenia with metabolic syndrome. The sample size is large and study span of 5 year have resulted in robust data. Over all study represent a major portion of Asian population. Despite these merits the study is not novel and is just an extension of earlier study by Kim S H et al., “Association between sarcopenia level and metabolic syndrome” (PMID: 33739984).

GENERAL

English should be improved throughout the text. Sentence needs to be rephrased and lines should be numbered throughout the manuscript.

In discussion include a paragraph explaining the parameters that are included in the current study and merit of the current study over earlier study conducted by same author (PMID: 33739984).

Methods and Materials

Methods and material section is well written

Results

The table 1 and table 2 are duplication of table 1 of previous published article (PMID: 33739984) by the same author represented in different way.

The figure 5B of the current study is duplication of figure 5 of previous published article (PMID: 33739984).

Figure legends should be self-explanatory.

Novelty

The study is not novel

6. PLOS authors have the option to publish the peer review history of their article (what does this mean?). If published, this will include your full peer review and any attached files.

Reviewer #1: No

Reviewer #2: No

---

## [Author Response · Author response to Decision Letter 0]

22 Jul 2021

Response Letter

Editors, PLOS ONE

Thank you for allowing the revision of our manuscript: ID PONE-D-21-15211 entitled “Association of obesity, visceral adiposity, and sarcopenia with an increased risk of metabolic syndrome: a retrospective study” We revised our manuscript in accordance with the reviewers’ suggestions. Our responses to the comments are as follows.

Reviewer #1: 

The manuscript provides evidence of the association between body composition parameters (obesity, visceral adiposity, and sarcopenia) and metabolic syndrome in a large sample of subjects. The manuscript is technically sound and the data do support the conclusions however there are a few revisions or clarifications I would suggest the authors address.

Comment 1>

1. The authors indicated that data from 4,621 repeated checkups were excluded from the analyses it would have been interesting to assess how the body composition parameters may have predicted changes in MS status over time using the repeated visits from the 4,621 with repeat data as this data was available.

Reply: Thank you for your valuable comments. It seems that causal relationship may be assessed by analyzing the data of subjects who underwent repeated health checkup. However, in this current study, we intended a cross-sectional study. We indeed have a plan to perform another study in the next step with the data of subjects who underwent repeated health checkup. As of now, it’s a pity that we can’t show you the repeat data, but we’ll be able to show you when the next paper is completed. We sincerely ask for your kind understanding. 

Comment 2>

2. The results are clear however, the authors should include the percentages for the total number of subjects (n=2238) with MS as well as the percentages for the total number of subjects who were in the obesity group, visceral adiposity group and sarcopenia group. I would further suggest either updating this in Table 1 and Table 2

Reply: Thank you for your valuable comments. Based on your comments, we added the percentages of subjects with MS in Table 1. We also added the percentages of subjects in the obesity group, visceral adiposity group and sarcopenia group in Table 2. 

Comment 3>

3. Table 3 and the language describing table 3 is unclear, it states the prevalence of MS was significantly higher in those with obesity, visceral adiposity AND sarcopenia compared to those without however, the table appears to show these as mutually exclusive groups. That is to say, obesity vs none, visceral vs non, sarcopenia vs none NOT those with obesity, visceral adiposity AND sarcopenia I would assume the group of individuals whom may be categorized by all 3 conditions would be much smaller. Thus, I would suggest the authors use ‘or’ rather than ‘and’ to indicate these analyses were done separately for each of the 3 conditions.

Reply: Thank you for your valuable comments. Based on your comments, we revised our manuscript. We used ‘or’ instead of ‘and’ to indicate these analyses were done separately for each of the 3 conditions. 

Comment 4>

4. It is unclear which models were used to assess the results reported in Table 3 and 4—were these done using chi-squared or the students t-tests? As these are counts for the number of parameters the authors should be more clear on how these results were modeled in the statistical analyses section.

Reply: Thank you for your valuable comments. We added more specific contents in the statistical analyses section. In the results section, we revised some contents regarding Table 3 and Table 4. 

Comment 5>

5. Table 5 is clearly defined and easy to read as well as reported and described appropriately both in the analyses section as well as the results.

Reply: Thank you for your valuable comments. 

Reviewer #2: 

The authors have undertaken study to explore the association of multiple body composition parameters such as obesity, visceral adiposity and sarcopenia with metabolic syndrome. The sample size is large and study span of 5 year have resulted in robust data. Over all study represent a major portion of Asian population. Despite these merits the study is not novel and is just an extension of earlier study by Kim S H et al., “Association between sarcopenia level and metabolic syndrome” (PMID: 33739984).

Comment 1>

1. English should be improved throughout the text. Sentence needs to be rephrased and lines should be numbered throughout the manuscript.

Reply: Thank you for your valuable comments. We rephrased some sentences and underwent additional English proofreading. And lines were numbered throughout the manuscript. 

Comment 2>

2. In discussion include a paragraph explaining the parameters that are included in the current study and merit of the current study over earlier study conducted by same author (PMID: 33739984).

Reply: Thank you for your valuable comments. As you suggested, we added sentences explaining the parameters that are included in the current study and merit of the current study over earlier study. We revised our manuscript as follows.

Discussion – 3rd paragraph

The previous study of the same study population as this study (Figure 1) showed that sarcopenia diagnosed by BIA is independently associated with MS risk in a dose-response manner.46 Our previous study focused on the relationship between sarcopenia and MS.46 In contrast, the current study comprehensively analyzed the effects of obesity, visceral adiposity, and sarcopenia on the risk of MS. Our current study demonstrated that the risks for MS additively increased……

Comment 3>

3. The table 1 and table 2 are duplication of table 1 of previous published article (PMID: 33739984) by the same author represented in different way.

Reply: Thank you for your valuable comments. Our current study and the previous published article share the same study population (N=13620), but the research topic and analysis methods were completely different between the two studies. However, since Table 1 and Table 2 in our current study have some overlap with the contents of the Table 1 of the previous paper, we added such contents in the main text and cited the previous paper in Table 1 and Table 2. 

Comment 4>

4. The figure 5B of the current study is duplication of figure 5 of previous published article (PMID: 33739984).

Reply: Thank you for your valuable comments. Although the research topic and analysis methods were completely different between the two studies, patients who performed both CT and Inbody on the same day were the same. Therefore, the graph that correlated CT and Inbody in this current study was the same as the graph in the previous study. We added such contents in the main text and cited the previous paper in the Figure 5B. 

Comment 5>

5. Figure legends should be self-explanatory.

Reply: Thank you for your valuable comments. Based on your comments, we revised the Figure legend of Figure 2. 

Comment 6>

6. The study is not novel.

Reply: Although this study is not novel, the sample size of our study is large and 5-year span resulted in robust data as you commented. And our data strongly support our conclusion. Thus, our study is thought to have sufficient merit. Thank you. 

Thank you again for your insightful advice.

Yours sincerely,

Ji Bong Jeong, MD, PhD 

Associate Professor

Department of Internal Medicine

Seoul Metropolitan Government Seoul National University Boramae Medical Center

20 Boramae-ro 5-gil, Dongjak-gu

Seoul 07061, Republic of Korea

Phone: +82-2-870-2222

Fax: +82-2-870-3863

E-mail: jibjeong@gmail.com

---

## [Decision Letter · Decision Letter 1]

2 Aug 2021

Association of obesity, visceral adiposity, and sarcopenia with an increased risk of metabolic syndrome: a retrospective study

PONE-D-21-15211R1

Dear Dr. Jeong,

We’re pleased to inform you that your manuscript has been judged scientifically suitable for publication and will be formally accepted for publication once it meets all outstanding technical requirements.

Kind regards,

Mauro Lombardo

Academic Editor

PLOS ONE

Additional Editor Comments (optional):

Reviewers' comments:

Reviewer's Responses to Questions

**Comments to the Author**

1. If the authors have adequately addressed your comments raised in a previous round of review and you feel that this manuscript is now acceptable for publication, you may indicate that here to bypass the “Comments to the Author” section, enter your conflict of interest statement in the “Confidential to Editor” section, and submit your "Accept" recommendation.

Reviewer #1: All comments have been addressed

Reviewer #2: All comments have been addressed

2. Is the manuscript technically sound, and do the data support the conclusions?

Reviewer #1: Yes

Reviewer #2: Yes

3. Has the statistical analysis been performed appropriately and rigorously? 

Reviewer #1: Yes

Reviewer #2: I Don't Know

4. Have the authors made all data underlying the findings in their manuscript fully available?

Reviewer #1: Yes

Reviewer #2: Yes

5. Is the manuscript presented in an intelligible fashion and written in standard English?

Reviewer #1: Yes

Reviewer #2: Yes

6. Review Comments to the Author

Reviewer #1: (No Response)

Reviewer #2: All the Major comments have been addressed.

Minor comments

1) The manuscript could be structured more reader friendly and improve the language by giving the manuscript for professionals for editing.

7. PLOS authors have the option to publish the peer review history of their article (what does this mean?). If published, this will include your full peer review and any attached files.

Reviewer #1: No

Reviewer #2: No

---

## [Editor Report · Acceptance letter]

5 Aug 2021

PONE-D-21-15211R1 

Association of obesity, visceral adiposity, and sarcopenia with an increased risk of metabolic syndrome: a retrospective study 

Dear Dr. Jeong:

I'm pleased to inform you that your manuscript has been deemed suitable for publication in PLOS ONE. Congratulations! Your manuscript is now with our production department. 

Kind regards, 

on behalf of

Dr. Mauro Lombardo 

%CORR_ED_EDITOR_ROLE%

PLOS ONE